# ITER Test Blanket Module—ALARA Investigations for Port Cell Pipe Forest Replacement

Jean-Pierre Friconneau [1,*], Tristan Batal [1], Olivier David [2], Chiara Di Paolo [3], Fabien Ferlay [1], Stéphane Gazzotti [1], Luciano Giancarli [3], Christophe Lacroix [1], Jean-Pierre Martins [3], Benjamin Michel [2] and Jean-Marcel Travere [1]

1   CEA-IRFM, F-13108 Saint-Paul-Lez-Durance, France
2   CEA, F-91190 Gif-sur-Yvette, France
3   ITER Organization Route de Vinon-sur-Verdon, CS 90 046, F-13067 St. Paul Lez Durance, CEDEX, France
*   Correspondence: jean-pierre.friconneau@cea.fr

**Abstract:** The objective of the ITER test blanket module (TBM) program is to provide experimental data on the performance of the breeding blankets in the integrated fusion nuclear environment. The ITER test blanket modules are installed and operated inside the vacuum vessel (VV) at the equatorial ports located within port plugs (PP), and each PP includes two TBMs. After each 18-month-long plasma operation campaign, the TBM research plan testing program requires the replacement of the TBMs with new ones during the ITER long-term shutdown, called long-term maintenance (LTM). The replacement of a TBM requires the removal/reinstallation of all test blanket system (TBS) equipment present in the port cell (PC), including those in the port interspace (PI), called pipe forest (PF). TBSs shall be designed so that occupational radiation exposure (ORE) can be as low as reasonably achievable (ALARA) over the life of the plant to follow the ITER policy. To implement ALARA process requirements, design activities shall consider careful integration investigations starting from the early phase to address all engineering aspects of the replacement sequence. The case study focuses on the PF replacement, in particular the port cell operations. This paper describes the investigations and findings of the ALARA optimisation process implementation in the early engineering phase of the PF.

**Keywords:** nuclear engineering; fusion engineering; ITER; test blanket system; health and safety; digital mock-up

## 1. Introduction: ITER Test Blanket Modules (TBMs)

Tritium breeding blankets (TBBs) ensuring tritium breeding self-sufficiency are a required feature for a demonstration power reactor (DEMO), the next step after ITER. Although a TBB is not required for ITER, since it will procure the tritium from external sources, it is included among the ITER missions that "ITER should test tritium breeding module concepts that would lead in a future reactor to tritium self-sufficiency, the extraction of high-grade heat and electricity production". All activities related to this mission correspond to the so-called "ITER TBM Program". A successful ITER TBM program represents an essential step for any fusion power development plan of all the seven ITER members.

The ITER TBM program [1], which will involve the manufacturing and testing of several different concepts [2] of the tritium-breeding module (Figure 1), is a key element of the fusion nuclear technology program. All ITER members contribute to the TBM program [3], and a testing strategy has been developed for approximately the first 10 years of ITER operation. Tests will be carried out on the systems, which consist of a TBM together with several ancillary systems providing independent cooling of the TBM, a dedicated tritium extraction system, a coolant purification system, and associated instrumentation and control (I&C) systems. A TBM and its ancillary systems are referred to as a test blanket system (TBS).

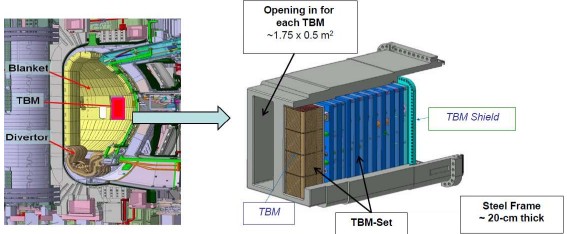

**Figure 1.** Test Blanket Module (TBM).

The TBSs must be fully integrated into the ITER facility. Moreover, it is required that operation of the TBSs should not affect ITER's operational performance, safety, or availability [3].

## 2. ITER Test Blanket System (TBS) Testing Plan

ITER represents the opportunity for testing breeding blanket mock-ups in a real fusion environment before the construction of the DEMO reactor. The TBS installation is planned for starting the operation at pre-fusion power operation 2 (PFPO-2) and continuing in the fusion power operation (FPO) phase. During the PFPO-2 phase, the effects of relevant magnetic fields, relevant plasma regimes (e.g., H-mode), surface heat fluxes, H permeation in the TBM First Wall coolant, and disruption-induced loads will be assessed. The PFPO-2 phase will also be essential for learning how to operate a TBS.

During the FPO phase, the effect of the relevant neutron flux, the volumetric heating, and tritium production, along with corresponding tritium extraction and transfers and integrated heat and tritium management capabilities, will be assessed.

Each TBS is functionally independent from the others. Two different TBSs will be tested in each of the two equatorial ports and, therefore, they will share the corresponding port plug (PP) and port cell (PC) area. The TBMs are the in-vessel part of the TBSs. In cases where a TBM-Set is not available, a dummy TBM will replace it.

This strategy, defined in the ITER research plan, implies that each TBM will have to be replaced several times (i.e., at each shutdown) requiring the removal and the re-installation of the TBS components present in the TBM PCs.

## 3. Configuration for the Test Blanket Systems (TBS)

The "ITER TBM Program" foresees the simultaneous operation of four TBMs, located in ITER equatorial ports 16 and 18 (Two TBMs per port). The TBMs are installed in a water-cooled steel frame (together with the associated shield) [4].

The TBM configuration planned for the first FPO test campaign is as follows:

At equatorial Port #16:

- Water-cooled lithium lead (WCLL) TBSs;
- Helium-cooled pebble bed (HCPB) or helium-cooled ceramic reflector (HCCR) TBSs.

At equatorial Port #18:

- Water-cooled ceramic breeder (WCCB) TBS;
- Helium-cooled ceramic breeder (HCCB) TBS.

Each TBM has its own ancillary systems (e.g., coolant, tritium extraction, instrumentation control, maintenance tools and equipment) located in the Tokamak Complex (TC). The TBM ancillary systems in the PCs are installed in a self-sustained steel structure called ancillary equipment unit (AEU).

The main associated ancillary systems forming the TBS (and not all present in each TBS testing configuration) are the following [5]:

- Primary cooling system, either helium (HCS) or pressurized water (WCS);
- Coolant purification system (CPS);
- Tritium extraction system (TES);

- Pb$^{16}$Li system as the tritium carrier;
- Tritium accountancy system (TAS);
- Instrumentation and control System (I&C);
- Neutron activation system (NAS), used to measure the neutron flux and fluency within the TBM.

The TBM port cell equipment provide the following functions [2]:

- To manage the TBS local processes in port cells, including, for example, two isolation valves per pipe in order to limit the effect of in-vessel and ex-vessel accidental cases and the local Pb16Li loop in the case of a liquid breeder.
- To provide the junction between the port plug mounted on the VV and the TBM feeding pipes transporting the cooling fluids, the tritium purge gas lines and also sample transfer lines for the neutron activation system (NAS) (measuring the local neutron flux and fluence within the TBM).
- To interface with the tokamak building and services through the common equipment, such as electrical power distribution, detritiation system for room ventilation, helium gas, cooling water, compressed air, instrumentation and control system (I&C) cabling, and vacuum detection services.

The bundle of pipes of the TBSs between the AEU and the PP is called the pipe forest (PF). These pipes are installed in a self-sustained steel structure part of the PF. This structure also supports the central bioshield plug.

Access in the port cells for personnel is possible through heavy nuclear doors, also called port cell doors (PCDs) and through personnel access doors (PADs), which are smaller doors inside the PCD that prevent opening the whole PCD for access. During transfer of large components such as PF or AEU, it will be necessary to open the large PCD. Both PCDs and PADs are designed to provide a shielding and confinement function.

During TBM removal operations, building gallery and PC areas are served by ventilation systems providing a pressure cascade. Opening PCDs or PADs are required to implement appropriate airlocks to maintain the pressure cascade personnel or component transfer.

## 4. Replacement Sequence of the ITER Test Blanket Modules (TBMs)

The TBM system requires scheduled replacement. Two dedicated port cells (#16 and #18) are allocated to the TBM program. The TBM exchange operation mainly concerns removal and reinstallation of the TBM port plug (TBB-PP).

The replacement of the TBM-PP is planned for each major long-term maintenance (LTM) shutdown every two years for an 8-month period, and it is achieved by means of the cask and plug remote handling system. In order to replace each TBM-PP, it is necessary to remove and subsequently reinstall most of the port cell components. To comply with this need, the PF and the AEU are designed as skid-mounted movable structures that have to be disconnected and transported to the appropriate hot cell maintenance areas. In the current plan, the AEU will be maintained during that replacement process, while the PF will be replaced by a new one in order to limit the overall exposure and contamination risks associated with the reconfigurations. The activated PFs will therefore be dismantled and disposed of as radioactive waste. In summary, to perform the TBM-PP replacement during each LTM shutdown, a full reconfiguration of the port cell (#16 and #18) is required.

The TBM port plug is attached onto the vacuum vessel (VV) port extension. The sealing flange provides the confinement barrier between the VV and PC areas. The TBM PC area contains two main zones: the port interspace (PI) and the remaining main port cell (see Figure 2). The separation is defined by the bioshield plug that provides continuity of the building bioshield. The PC door to the building gallery provides the required segregation between the two different nuclear ventilation zones. All permeating surfaces of the PF and AEU are enclosed in shrouds connected to the detritiation system, providing a suitable pressure cascade to meet the port cells' nuclear ventilation requirements.

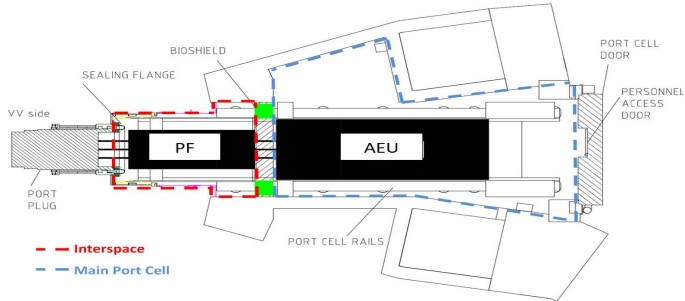

**Figure 2.** TBM port cell #16 configuration.

During the replacement operation, when plasma is off, the gamma radiation is therefore the result of the decay of activated materials. The level of activation of the TBM-PP [6] prevents direct exposure of the worker to the front part of the TBM (the one facing the plasma).

Dust will be generated inside the ITER vacuum vessel (VV) by interaction of the plasma with the facing components during steady-state operation and transients (mainly disruptions). Dust on the TBM-PP will have the same radiological characteristics as the first few microns of the plasma-facing components and will thus be both activated and tritiated. Under those radiological hazard conditions, the TBM-PP replacement operation, the last step of the sequence described in Figure 3, is performed by a fully remote handling operation [7]. The removal and transportation of the in-vessel component TBM-PP is performed by the cask and plug remote handling system (CPRHS). During CPRHS operations, human access in the port cell is not allowed. It is therefore required to prepare the fully remote CPRHS operation by providing the necessary clearance in the port cells (remove PF and AEU). In addition, because the CPRHS travels between the PC area and the building gallery, the PC area's radiological conditions in terms of contamination shall be aligned (i.e., airborne and surface contamination). In summary, the requirement to perform TBM-PP removal (see step 3 in Figure 3) drives the performance for achieving the other steps in terms of space released and radiological conditions.

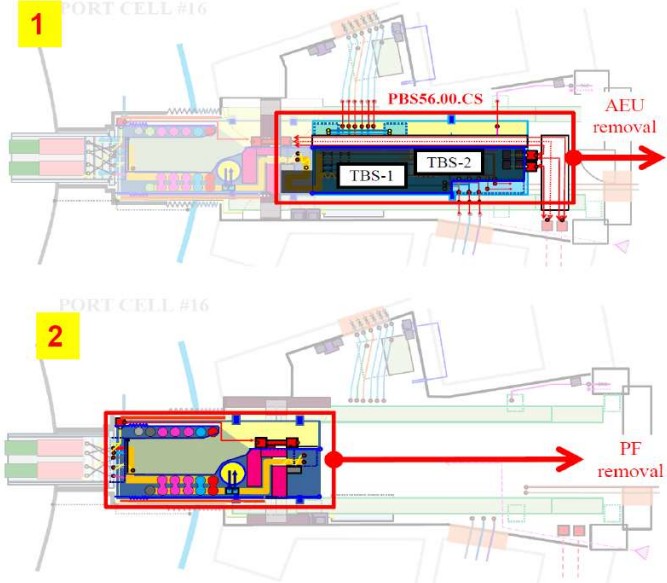

**Figure 3.** *Cont*.

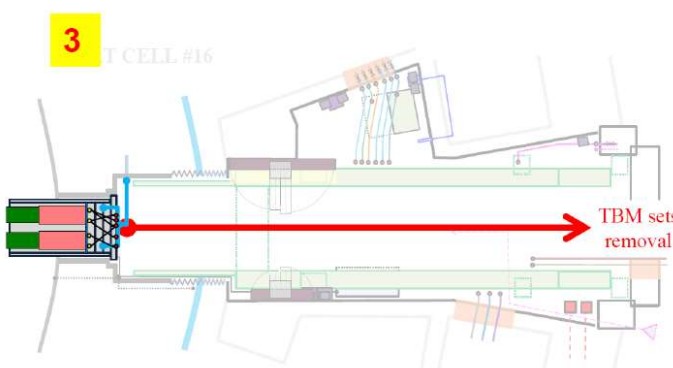

**Figure 3.** TBM PC removal sequence (3 steps).

Steps one and two are performed when the TBM-PP is on the VV. The TBM-PP in the place provides proper shielding to allow human access in the PC areas to the PF and AEU to perform replacement operations.

During step 1 and step 2, the TBP-PP is sealed onto the VV, which prevents worker exposure to the VV contamination radiological conditions.

In summary, hands-on operations in the port cells (#16 and #18) required during the replacement of PF and AEU will induce occupational radiation exposure (ORE).

The PF shall be removed to enable TBM PP exchange, as shown in the second step in Figure 3. The PF removal, requires access to the area between the PP and the PF to release/install the pipe junctions. This area is congested with many components, which increase the difficulty for human access and circulation in working conditions compliant with ITER rules. In addition, the working area to access pipe junctions is located in the port interspace (PI) area of the PC (Figure 2) beyond the bioshield plug, and therefore has highest radiological hazard.

The PC component removal task is the most critical as it occurs immediately after plasma operation. In normal conditions, the dose rate in the PC should be very similar during installation and removal operations [6]. Specific radiological hazards shall be considered during the removal steps. The removal sequence assumes disassembly of PF after plasma campaign and therefore might expose the worker to volumes where activated dust or tritium have been trapped. On the other hand, installation of PF considers and assumes a new, free from contamination PF component upon each replacement. Because of those conditions, the design of PF requires consideration of the full life cycle of the system, including the removal and re-assembly of the PF, where the ALARA process shall be implemented starting from the early phase of the design.

The standards and guidelines for radiological safety developed for ITER are based upon international standards and recommendations from the International Atomic Energy Agency (IAEA) and the International Commission on Radiological Protection (ICRP).

## 5. ALARA for TBS Replacement Operations

The TBM system should be designed such that occupational radiation exposure can be as low as reasonably achievable (ALARA) over the lifetime of the plant. Before removing/installing the PF (and AEU), all junctions should be released/connected. The activity to perform port cell equipment junction assembly/disassembly should comply with the need to minimise doses for occupational exposures using the recommendations from the International Atomic Energy Agency [8].

Provisions for dose reduction measures (DRM) for radiation in working areas and a reduction in occupancy times in radiation fields are means to reduce operator exposure.

The ALARA approach being an iterative process, this paper highlights the necessity/advantages of applying it to the replacement sequence in the early design phases of the TBSs so that it leads to an overall/substantial ORE reduction.

The implementation of the ALARA requirement [9] implies considering optimization process since the preliminary design phase. The process scope is to identify and implement the proper means that minimise worker radiation exposure. Means to reduce personal dose exposure [8]. are of two categories:

a. **Reduction in dose rates** in working areas by the following:

- Source reduction;
- Improvement in shielding (e.g., local shielding);
- Increasing the distance between workers and sources;
- Ensuring good ventilation (e.g., flow rate, air renewal, etc.).

b. **Reduction in occupancy times** in radiation fields by the following:

- Specifying high standards for equipment to ensure very low failure rates, (e.g., equipment quality class, etc.);
- Ensuring ease of maintenance or ease of removal of equipment;
- Simplifying operating procedures;
- Ensuring ease of access and good lighting (e.g., ergonomics, etc.).

An initial list has been identified for equipment and tools for their potential benefits during operation in the port cells:

- Human access equip. (scaffolding, stairs, etc.);
- Temporary lifting devices (floor-based);
- Visual equip. (cameras, lights, etc.);
- Detectors;
- Personal protection equipment;
- Temporary local shielding (lead bricks, etc.);
- Decontamination equipment;
- Radwaste handling/storage equipment.

Another way to group the provisions for minimizing doses is to differentiate protection against external exposure and protection against internal exposure.

- Provisions against external exposure:
    - Radiological zoning;
    - Radiation shields;
    - Choice of materials;
    - Dose optimization through design changes;
    - Remote handling and maintenance;
    - Access control systems;
    - Minimizing occupational exposure times;
    - Limiting surface and atmospheric contamination;
    - Monitoring systems.

- Provisions against internal exposure:
    - Collective protective means;
    - Static confinement;
    - Dynamic confinement;
    - Personal protective equipment (PPE).

The steps in the ALARA process for the design of the PF shall follow three general principles of radiation protection, as defined in [8]. These concern justification, optimization of protection, and application of dose limits (limitation of doses). Applied to the PF design, those three aspects translate into the following statements:

- **Justification**: The activity to replace the pipe forest is required to implement the ITER test blanket systems (TBS) as per the ITER research plan (see Section 1).
- **Optimization**: "Protection is optimized" means that optimization of protection has been applied and the result of that process has been implemented:
    - The collective dose will be used as an instrument for optimization;

- ORE initial assessment will identify the highest priority area of optimization;
- Iterative optimization steps will allow the reduction in dose personal exposure by means of available radiological technologies and protection procedures called dose reduction measures (DRM).

- **Dose limitation**: This shall comply with any relevant dose limit from project guidelines on occupational exposure [10]:

- Individual worker dose rate of 0.1 mSv/h;
- Annual individual worker dose of 10 mSv/y;
- Annual individual average dose of 2.5 mSv/y (all workers);
- Collective annual worker dose of 500 mSv/y.

## 6. TBSs in PC Replacement Sequence: Initial ORE in FPO

In the current stage of the PF design, the first ALARA optimisation step is to perform an initial assessment of the ORE associated with the replacement of the PF. In practice, upon the design maturity of the system, there is a very large uncertainty regarding the occupancy time. For example, work in an air suit compared to work without an air suit with a skilled human operator (unencumbered) significantly increases occupancy time (8:1 as per ref. [11]). The initial ORE for replacement of TBSs in the port cell is quoted with relative values (see Table 1).

**Table 1.** Initial assessment of the ORE for PC activities during TBM replacement.

| Workstations | Occupancy Time% | Occupancy Dose% |
|---|---|---|
| # 1 (in Gallery) | 37 | 2 |
| # 2 (in PC area) | 6 | 2 |
| # 3 (in PC area) | 8 | 3 |
| # 4 (in PC area) | 23 | 20 |
| # 5 (in PI area) | 2 | 3 |
| # 6 (in PI area) | 24 | 70 |

A workstation is a place where work of a particular nature is carried out. To support the ALARA study in port cell #16, six workstations have been identified (see Figure 4). In the recent neutronics analysis, it was found that the shutdown dose rate (SDDR) from activated components due to plasma neutrons at 12 days after shutdown is up to 100 μSv/h [6] in workstations #5 and #6 of the port cell #16 and lower in the other workstations. This ORE initial assessment enables us to identify the highest priority area of optimization. Table 1 shows the results of the first ORE assessment of the sequence to remove and reinstall the AEU and the PF.

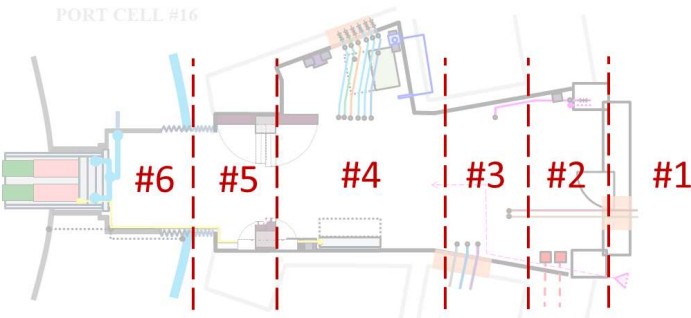

**Figure 4.** Workstation identification for PC#16.

Table 1 provides occupancy time distribution of the collective time spent by workers and occupancy dose distribution for the collective dose of workers across the workstations, listed in rows.

Table 1 shows that workstation #6, located in the PI, contributes to two-thirds of the occupancy dose in PC #16. The outcome is that to reduce ORE, DRM shall focus as a priority on the activity for the PF replacement, removal, and installation.

Another result relevant for shutdown management is that most of the time spent for the TBM replacement operation is in workstation #1, located in the gallery of the building outside the port cell #16. In practice, this area is providing the integrated logistics services required for the replacement operation in the port cell. The management of the other activities in the gallery during shutdown should be carefully oversighted and managed.

## 7. Reduction in Occupancy: PF Design Opportunity

During the PF design engineering activities, two key design requirements were apparently conflicting:

- Piping layout should comply with the RCC-MRx code of construction [12];
- Piping layout should accommodate the required clearance for human circulation corridors to allow access to workstations (Figure 5).

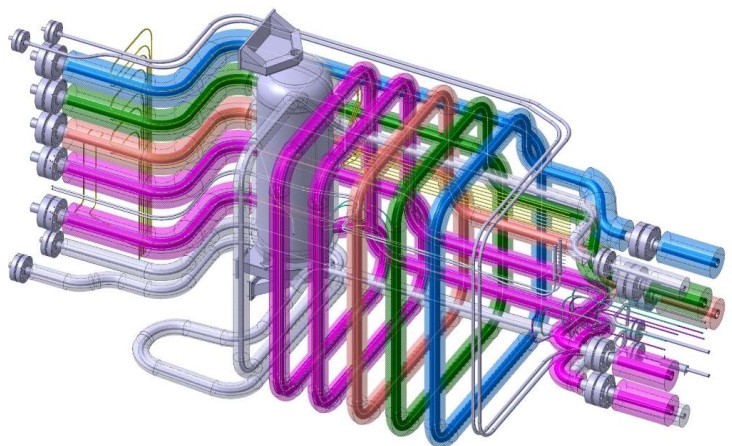

**Figure 5.** PF design layout in optimised configuration.

To satisfy RCC-MRx design criteria, the initial PF layout required design changes and introduced a reduction in clearance for human circulation. To satisfy the conditions for human circulation for operation in air suits, increasing the size of the corridors in PI areas is also required. Both requirements introduce conflicting engineering parameters for clearance in PI areas. Available space in the PI area is then frozen by building limits.

This conflict was solved by providing extra space for the piping routing in the PF radial directions (Figure 5). This introduced the necessary additional pipe length and flexibility to comply with the RCC-MRx design criteria. In addition, large clearance was released in the PF central area required for human circulation corridors to access workstations.

This demonstrates that considering the ORE requirement from the early design phase is a good opportunity to secure and stabilize the main design options as soon as possible.

To secure the PF design layout as much as possible, investigations on PF digital mock-up have been performed [13]. Workstation #6 is where the operator needs to work on the PF junction close to the PP (Figure 6). This area is where SDDR is the highest in the PC [6], and workstation clearance is not free from pipes.

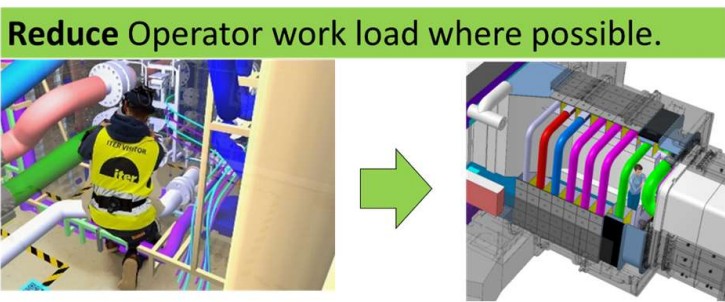

**Figure 6.** DRM increases human access.

The priority is workstation #6 (Figure 4) for investigating the digital mock-up's working conditions. The outcome provides guidelines to enhance the piping layout to optimise operator access conditions.

## 8. Reduction in Occupancy: PF Interface Opportunity

The hierarchy of control, selected by the ITER organisation, used to define the DRMs is based on the ERIC-PD approach. ERIC-PD stands for eliminate, reduce, isolate, control, personal protective equipment, and discipline.

The most efficient level of DRM relies on the "Elimination" level. Along the PF replacement sequence, the removal of complex tasks is therefore investigated. Installation of the PF component requires operations on about 30 pipe junctions at each campaign. A typical sequence of installation of LiPb PF pipe welded junction is as follows:

1. Pipe end preparation;
2. Pipe alignment;
3. In-pipe inerting;
4. Pipe pre-welding heat treatment;
5. Pipe welding;
6. Pipe visual inspection;
7. Pipe penetrant test (PT);
8. Pipe radiographic test (RT);
9. Pipe post-welding heat treatment;
10. Connect pipe heating wires;
11. Install pipe thermal insulation.

When considering flanged junctions, pipe inerting and pipe heating steps are not required, and pipe junction tests are simplified.

Therefore, where possible [14], the opportunity to implement the flanged junction option should be considered for its high-efficiency DRM (eliminate), in particular when considering the large quantity of junctions at the PF (Figure 7).

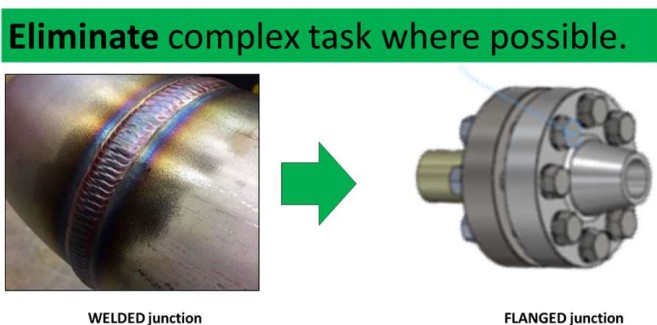

**Figure 7.** DRM: remove complex task.

### 9. ALARA and ORE Iterative Improvement Process

ALARA is an iterative optimisation process used to reduce personal dose exposure by means of available radiological technologies and protection procedures called dose reduction measures (DRM).

In the next steps of ORE investigation, the following provisions (defined in general safety guidelines [8]) should be considered for their benefits to reduce the ORE during TBM replacement operation. This will lead to the following aspects:

- Shielding:
    - Additional local shielding should be provided to reduce the radiation field as needed.

- Ventilation: In many facilities, the control of airborne contamination is achieved by
    - Maintaining adequate negative pressure with respect to atmospheric pressure;
    - Providing an adequate or prescribed number of air changes in the workplace;
    - Providing appropriate exhaust air, so that the discharges from the facility will be within authorized limits.

- Dust control (risk of dust release during the disassembly process and/or dry processing of radioactive material).
    - The generation of dust in operations should be reduced to the extent practicable by the use of appropriate techniques;
    - Where dust is generated, it should be suppressed at the source. Where necessary and practicable, the source should be enclosed under negative air pressure;
    - Dust that has not been suppressed at the source may be diluted to acceptable levels by means of frequent changes in air in the working area;
    - Care should be taken to avoid the resuspension of dust as a result of high air velocities;
    - Where methods of dust control do not achieve acceptable air quality in working areas, enclosed operating booths with filtered air supplies should be provided for the workers.

- Spillage of radioactive material operating procedures to be followed in the event of any significant radiation hazard or potential radiation hazard arising from the spillage of radioactive material.
    - Cleaning up spillages: The area should be decontaminated by the removal of all loose contamination and contaminated materials to the greatest extent practicable;
    - Restricting access to the area.

- Surface contamination work with unsealed radioactive substances creates the potential for contamination of surfaces. The physical design features used for contamination control may include the following:
    - Specific design features aimed at confining radioactive material to prevent it from causing surface contamination;
    - Ventilation systems aimed at preventing the build-up of surface contamination as a result of the settling of airborne particles;
    - Monitoring for surface contamination;
    - Housekeeping.

- Decontamination of equipment and decontamination of personnel
    - Decontamination of equipment and areas of floors and walls;
    - Decontamination of personnel.

### 10. Conclusions

This developed case study focuses on pipe forest replacement operations in the PC. Investigations to implement the ALARA optimisation process in the early engineering phase of PF have been successful. The optimisation process has been tested on a relevant

case and results have been incorporated into the PF component design. However, further iterations are required to improve the evaluation of the ORE considering the current engineering maturity:

- AEU and PF systems' design maturity is low;
- Sequence definition maturity (steps and duration) is low;
- There is high uncertainty regarding occupancy time (~1:8), highly reliant on working conditions (work in suits, workstation configuration, etc.).

## 11. Perspectives

This investigation has initiated activities to further develop DRMs. In the future, the validation of flanged junctions' feasibility for the various pipes' configuration would be an opportunity for a substantial ORE reduction. Another line of investigation concerns the reduction in occupancy time by means of robotics and cobotics technology [15,16]. Finally, during the early stages of design, the use of digital mock-up for the validation steps of the replacement operation will be useful to cope with the PF and AEU design maturity and associated evolutions.

## 12. Disclaimer

The views and opinions expressed herein do not necessarily reflect those of the ITER organisation.

**Author Contributions:** J.-P.F.: Investigation, writing—original draft preparation, writing—review and editing. T.B.: writing—review and editing. O.D.: writing—review and editing. C.D.P.: writing—review and editing. F.F.: writing—review and editing. S.G.: writing—review and editing. L.G.: writing—review and editing. C.L.: writing—review and editing. J.-P.M.: writing—review and editing. B.M.: writing—review and editing. J.-M.T.: writing—review and editing. All authors have read and agreed to the published version of the manuscript.

**Funding:** This research received no external funding.

**Data Availability Statement:** Not applicable.

**Conflicts of Interest:** The authors declare no conflict of interest.

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
