# Peer review of "ITER Test Blanket Module—ALARA Investigations for Port Cell Pipe Forest Replacement"

_jne, doi:10.3390/jne4010022_

Round 1
Reviewer 1 Report
1. take out the references from the abstract
2. What is a Dummy TBM? Where is it located? This dummy is an alternative for the TBM-SET???
3. What are the Specific radiological hazard considered for the removal? the authors?
4. Figure 6 is not clear, please increase resolution
5. What is the gallery (WS#1)??
6. What does table 1 indicates? Inititally I have the occupancy time of the different area in %, but what does the authors refers with the occupancy dose? It is no clear how does the author reach this values. It should be clear what the parameters considered.
Also, table 1 is the only analysis showed, what is the approximation of the total dose per year for the OREs?
7. the author states "Piping layout shall accommodate with the required clearance for human circulation corridor to allow access to workstations (Fig.6)."
The author mention the figure 6 as it could be possible to see the human circulation corridor to allow access workstation, but it is not clear for the reader what is the corridor mentioned.
8. Conclusions should not be state include conclusions section, Because, the conclusion should be from the work of the authors, and It should not reference other works
9. This work is interesting but I expected to see the dose from the workers with occupational radiation exposure. As this work seems an application of ALARA statement, I felt a lack of the differences between the procudes implemented to reduce the time and occupational exposure of the workers and procedures without applying the ALARA statement.
Author Response
Thank you for your kind review
1 References removed from the abstract
2 - "In case a TBM-Set is not available it will be replaced by a Dummy TBM": This means : Dummy TBM replaces the TBM-Set to provide the necessary function (like shielding) to run the ITER machine. Dummy TBM shall be installed to allow re-start of the plasma operations in case TBM-set is not ready for installation.
3 - Removal sequence assumes disassembly of PF after Plasma campaign and therefore might expose the worker to volumes where activated dust or Tritium has been trapped. This is specific to Diassembly steps.
- The paper has been revised to clarify this point.
4 - Figure 6 has been updated to improve resolution
5 - A workstation is a place where work of a particular nature is carried out. To support the ALARA study in port cell #16, 6 workstations have been identified (see Fig.6).The gallery (WS#1) is the workstation #1 in located outside the PC#16 called the gallery.
- The paper has been revised to clarify this point.
6. Table 1 provides Occupancy time distribution of the collective time spent by workers and Occupancy dose distribution for the collective dose of workers across the workstations listed in rows.
- The paper has been revised to clarify this point.
Collective annual worker dose of 500 mSv/y for the facility is available in the paper, the PC#16 specific values is not. At that stage of the design maturity, current assessment focuses relative values (in %) between Workstations to define priorities in the ALARA optimisation process.
7- figure 7 has been updated to better illustrate the clearance provided at the center of the pipe forest arrangement required for man access.
8- Perspectives and other work has been removed from the conclusion of the revised paper
9- Thank you for your kind comment. In the paper, only ORE initial assessment to identify the highest priority area of optimization is described in this paper. The impact of DRM on the ORE is not yet available, and will be assessed when PF design will be stabilized to verify efficiency of the various DRM's.
As you pointed perfectly, the efficiency of the DRM's is not evaluated yet, but this paper shows how ALARA implemented very early in the design phase can provide guidelines to the design process to include "maintenance" aspects.

Reviewer 2 Report
This paper describes investigations and findings of the ALARA optimisation process implementation in the early engineering phase of the PF.
1. The main contribution of the paper should be highlighted and emphasized. It would be great if the drawbacks and gaps of literature are clear and, particularly, how the proposed approach aims at filling these gaps.
2. The abstract should briefly display the results of the research.
3. It is better to make a more comprehensive literature review in the form of a table (matrix) so that the reader is more confident with the contribution of this research.
4. The performance required presenting in more quantitative manner
5. Improve the English writing/editing of the manuscript. There are many grammatical mistakes throughout the manuscript.
6. Add more results to validate the proposed work and compared those with the existing analysis/work. Moreover, the computational effort and accuracy of the proposed work should be compared with a benchmark method and other existing work to justify its effectiveness.
7. Explain in brief how the present paper differs from the published ones.
8. Present the proof of sensitivity and robustness formulations/analysis of the proposed work and validation.
9. It is necessary that the authors should illustrate/present the details of the proposed work, modeling/design and data of the studied power system, system constraints/data/parameters, etc. Moreover, state the system constraints, in other words, the upper and the lower boundaries of the optimization algorithm/system variables, etc.
10. What are the limitations and disadvantages of the proposed work?
Author Response
Thank you for your kind contribution in the review of this paper. Please find some clarifications in reply to the list of comments/recommendations provided:
- The “As Low As Reasonably Achievable” (ALARA) principle is largely followed during operation or decommissioning of nuclear facilities. Implementation of ALARA principle in the early engineering phase of a facility is not documented in the literature in general and in ITER in particular.
- This paper provides engineering feedback on implementation of ALARA principle in the design of a complex ITER system that requires frequent replacement operations.
- The available references are distributed along the text to provide in context the required elements. No similar paper has been found in the Fusion Engineering and Design domain and associated literature.
- At the early stage of the engineering phase, the performances are only available at qualitative level. This paper aims to illustrate that qualitative element provides relevant inputs to contribute to the design process.
- Best effort will be provides to improve the quality of writing and editing in the revised paper.
- The paper is about an engineering investigation, engineering and qualitative results are exposed.
- Paper published on ALARA reports on results and benefits for facility operation and/or decommissioning phase. This paper reports on results and benefits for facility under design phase (ITER case). The specific case of the TBM system described in the paper is even in a very early phase of the design (no Equivalent paper found yet).
- Not applicable for the engineering work presented.
- The scope of the paper stands for engineering investigation, ie implementation of the ALARA process in the early phase of the design for a specific ITER system: the TBM. This paper should provide confidence that ALARA principle can be introduced since early phase in the design process when quantitative set of data are not yet available.
Thank you again for your time in the review process. I hope this reply provides enough clarification to complete the review process.
Kind regards.
Round 2
Reviewer 1 Report
This work is about the occupational radiation exposure for the test blanket module program at ITER.
The calculations are important for the ITER Test Blanket Modules, but theres is no calculation about the ORE, maybe a monte carlo simulation would be nice to add to the present work and fill the occupancy dose not only with percentage dose but also with a calculation of the dose for each workstation.
How was performed the neutronic analysis?
Author Response
Thank you very much for your contribution in this review.
This work is about the ALARA as optimisation criteria for the TBM design process. This work is not reporting about the neutronic analysis.
The relevant neutronics analysis values are available at the following reference of the paper:
- Harb & al. «Neutronics analysis and assessment of shielding options of pipe forest and Bioshield-Plug design for ITER TBSs », Fusion Engineering and Design, 2021
Because of the lack of uncertainty due to the current ITER of design maturity, only ORE relative values are exposed (ie %) in this paper.
This paper shows that this approach to use relative values is sufficient to drive the TBM design process.
I hope this clarifies your questions on the current paper. Enclosed a revised version of the paper with expanded description on the methodology.
Reviewer 2 Report
The authors have carefully revised the article and I think it is acceptable.
Author Response
Thank you for your kind review, enclosed an expanded version of the paper to match the requirement of 4000 words.
